# Alcohol-Induced Oxidative Stress and Gut–Liver–Brain Crosstalk: Expanding the Paradigm from ALD to MetALD

**DOI:** 10.3390/antiox14101196

**Published:** 2025-10-01

**Authors:** Jeong-Yoon Lee, Young-Min Jee, Keungmo Yang, Tom Ryu

**Affiliations:** 1Department of Neurology, Soonchunhyang University College of Medicine, Seoul 04401, Republic of Korea; ljy890818@schmc.ac.kr; 2Department of Translational Medicine, Graduate School of Medicine, Seoul National University College of Medicine, Seoul 03080, Republic of Korea; 3Department of Family Medicine, Soonchunhyang University College of Medicine, Seoul 04401, Republic of Korea; ymjee27@gmail.com; 4Department of Family Medicine, Graduate School of Medicine, Seoul National University College of Medicine, Seoul 08826, Republic of Korea; 5Department of Internal Medicine, Division of Gastroenterology and Hepatology, College of Medicine, The Catholic University of Korea, Seoul 06591, Republic of Korea; 6Department of Internal Medicine, Institute for Digestive Research, Digestive Disease Center, Soonchunhyang University College of Medicine, Seoul 04401, Republic of Korea

**Keywords:** alcohol, gut-liver-brain axis, neuroinflammation

## Abstract

Alcohol-associated liver disease (ALD) includes a spectrum from steatosis and steatohepatitis to cirrhosis and hepatocellular carcinoma driven by oxidative stress, immune activation, and systemic inflammation. Ethanol metabolism through alcohol dehydrogenase, aldehyde dehydrogenase, and cytochrome P450 2E1 generates reactive oxygen and nitrogen species, leading to mitochondrial dysfunction, hepatocellular injury, and activation of inflammatory and fibrogenic pathways. Beyond hepatic effects, ALD engages the gut–liver–brain axis, where microbial dysbiosis, blood–brain barrier disruption, and neuroinflammation contribute to cognitive impairment and cerebrovascular risk. The emerging concept, metabolic dysfunction-associated steatotic liver disease and increased alcohol intake (MetALD), presents the synergistic impact of alcohol and metabolic comorbidities, enhancing oxidative injury and fibrosis. This review summarizes key mechanisms connecting oxidative stress to multisystem pathology and highlights the need for precision therapies targeting redox imbalance, immune dysregulation, and gut–brain–liver interactions to improve outcomes in ALD and MetALD.

## 1. Introduction

Alcohol is one of the most widely consumed psychoactive substances worldwide, deeply ingrained in various cultures and social practices. While moderate consumption might be used for short-term mood elevation and social relaxation effects, chronic and excessive alcohol intake remains a major global health concern. Alcohol-associated liver disease (ALD) includes a clinical spectrum ranging from simple steatosis and steatohepatitis to advanced fibrosis, cirrhosis, and hepatocellular carcinoma (HCC) [1]. Traditionally, ALD has been defined by quantity and duration of alcohol intake, with clinically significant alcohol consumption generally considered as ≥210 g per week for men and ≥140 g per week for women [2]. However, recent findings indicate that this framework inadequately captures the complex, multifactorial nature of alcohol-related liver injury [3].

A key mechanism underlying alcohol-related liver injury is oxidative stress [4]. Highly reactive molecules, such as reactive oxygen species (ROS) and reactive nitrogen species (RNS), affect the antioxidant defense systems in hepatocytes, resulting in lipid peroxidation, mitochondrial dysfunction, protein misfolding, and DNA damage [5]. In parallel, alcohol-induced oxidative stress activates inflammatory pathways, including nuclear factor-kappa B (NF-κB), inflammasomes, and cytokine release, further aggravating hepatic injury and fibrosis [6].

Beyond hepatic consequences, alcohol affects multiple organs and systems, including the central nervous system. Recent studies have highlighted hepatic glutamate signaling as hepatic stellate cell (HSC) activation and natural killer cell modulation, which might indirectly contribute to systemic inflammation and brain–liver crosstalk [7,8]. Chronic alcohol consumption has been shown to dysregulate glutamate signaling at the level of both hepatocytes and hepatic non-parenchymal cells, suggesting the existence of a metabolic synapse between the liver and neural networks [7]. Moreover, the effects of alcohol on the gut-brain axis, via alterations in intestinal permeability, microbial dysbiosis, and endotoxemia, establish a systemic inflammatory microenvironment that impacts both hepatic and neurologic function [9].

Meanwhile, the emergence of metabolic dysfunction-associated steatotic liver disease and increased alcohol intake (MetALD) represents a paradigm shift in the conceptualization of alcohol-related liver pathology. Unlike traditional ALD, MetALD integrates the role of metabolic risk factors such as obesity, insulin resistance, and dyslipidemia in initiating alcohol-related liver injury [10]. This evolving framework correlates closely with the recent nomenclature changes in metabolic liver disease, including metabolic dysfunction-associated steatotic liver disease (MASLD), and highlights the overlapping pathways of oxidative stress, immune dysregulation, and metabolic imbalance [11,12].

ALD refers to steatotic liver disease driven primarily by harmful alcohol use (defined as >60 g/day in men or >50–60 g/day in women, equivalent to >420 g/week and >350 g/week, respectively). MASLD refers to steatosis in the presence of metabolic dysfunction when alcohol use is below harmful thresholds (≤210 g/week for men and ≤140 g/week for women). MetALD describes patients meeting MASLD criteria while consuming alcohol above these MASLD thresholds but below harmful ranges (210–420 g/week for men; 140–350 g/week for women), reflecting a dual burden of metabolic dysfunction and alcohol use [11].

In this review, we aim to provide a comprehensive overview of alcohol-induced oxidative stress and its systemic consequences, including its role in the pathogenesis of liver injury, immune activation, neurologic signaling, and cerebrovascular disease. We further explore the intersection between ALD and metabolic dysfunction through the perspectives of MetALD, and discuss emerging therapeutic strategies targeting oxidative and inflammatory pathways across organ systems. By integrating recent insights into the gut-liver-brain axis and neurologic-liver crosstalk, this review offers several perspectives on the evolving pathophysiology and future directions in alcohol-related liver disease.

## 2. Alcohol Metabolism and Liver Pathophysiology

### 2.1. Overview of Ethanol Metabolism in the Liver

Ethanol metabolism in the liver is mediated through three primary enzymatic pathways: alcohol dehydrogenase (ADH), acetaldehyde dehydrogenase (ALDH), and the microsomal ethanol-oxidizing system (MEOS), which involves cytochrome P450 2E1 (CYP2E1) [13]. In the cytosol, ADH catalyzes the oxidation of ethanol to acetaldehyde, which is then converted to acetate in the mitochondria via ALDH [14]. Acetate is further metabolized to carbon dioxide and water or utilized in peripheral tissues for energy production [15]. Under conditions of chronic alcohol intake, the MEOS pathway becomes increasingly important. The induction of CYP2E1 not only enhances ethanol clearance but also produces a substantial amount of ROS, contributing to oxidative stress and liver injury [16].

Acetaldehyde, a key intermediate in ethanol metabolism, is both hepatotoxic and immunogenic [17]. It forms covalent adducts with cellular macromolecules, including proteins, lipids, and DNA. These adducts interfere with normal cellular function, disrupt protein folding, and could lead to the activation of innate immune responses [18]. Furthermore, acetaldehyde promotes mitochondrial damage and sensitizes hepatocytes to other insults such as lipotoxicity and inflammation [19].

### 2.2. Generation of Oxidative Stress and Redox Imbalance

A hallmark of alcohol metabolism is the generation of ROS and RNS, particularly through the CYP2E1 pathway [20]. ROS such as superoxide anion, hydrogen peroxide, and hydroxyl radicals induce direct oxidative damage to cellular membranes via lipid peroxidation [21]. They also oxidize mitochondrial DNA, proteins, and other macromolecules [22]. Additionally, ROS disrupt the mitochondrial respiratory chain, impairing adenosine triphosphate production and leading to further ROS generation, establishing a vicious cycle of oxidative damage [23].

Concomitantly, ethanol metabolism alters the cellular nicotinamide adenine dinucleotide (NAD)^+^/NADH ratio, favoring a highly reduced intracellular state. This metabolic shift impairs fatty acid β-oxidation and enhances de novo lipogenesis, contributing to hepatic steatosis [24]. The depletion of NAD^+^ also affects the activity of key enzymes involved in DNA repair and mitochondrial function, such as sirtuins and poly (ADP-ribose) polymerases (PARP), exacerbating cellular vulnerability to oxidative stress [25].

### 2.3. Immune Activation and Inflammatory Pathways

While oxidative stress is a key initiator of alcohol-induced hepatocellular injury, immune activation and chronic inflammation are critical inducers that determine the progression and severity of liver disease. Alcohol consumption disrupts gut barrier function, facilitating the translocation of microbial products, such as lipopolysaccharide (LPS), into the portal circulation. These gut-derived pathogen-associated molecular patterns engage pattern recognition receptors (PAMPs), particularly Toll-like receptor 4 (TLR4), on Kupffer cells, the resident hepatic macrophages [26]. Activation of TLR4 triggers downstream signaling pathways including nuclear factor-κB (NF-κB) and inflammasomes, leading to the production of proinflammatory cytokines such as tumor necrosis factor-α (TNF- α), interleukin (IL)-1β, and IL-6 [27].

In the ALD, hepatocellular damage also induces the release of damage-associated molecular patterns (DAMPs), including high-mobility group box 1 and mitochondrial DNA. These endogenous danger signals further activate Kupffer cells and infiltrating immune cells, sustaining a state of sterile inflammation [28]. Neutrophils are rapidly recruited to sites of injury via chemokine gradients and contribute to liver damage through the release of proteases, ROS, and neutrophil extracellular traps [29].

This inflammatory cascade also involves hepatic stellate cells (HSCs), which are activated in response to inflammatory cytokines and oxidative signals. Once activated, HSCs transdifferentiate into myofibroblasts and secrete extracellular matrix proteins, leading to hepatic fibrosis. Key fibrogenic mediators include transforming growth factor-β and IL-17, the latter primarily secreted by pro-inflammatory T helper 17 (Th17) cells [30]. These adaptive immune responses are notably altered in ALD, with Th17 cells expanded and regulatory T (Treg) cells functionally impaired, creating an immune environment that accelerates inflammation and inhibits resolution [31]. Recent work has demonstrated that alcohol-induced mitochondrial double-stranded RNA could be exported via exosomes to activate TLR3 in Kupffer cells, thereby enhancing IL-1β expression and IL-17A production, further enhancing inflammatory cascades in ALD [32].

The immunologic signature of ALD overlaps significantly with metabolic liver diseases. In MetALD, metabolic endotoxemia, adipokine dysregulation, and insulin resistance further enhance immune activation [33]. Adipose-derived mediators including leptin, adiponectin, resistin, and other cytokines/chemokines modulate Kupffer cell priming, HSC activation, and systemic inflammation [34]. The dual insult of alcohol and metabolic dysfunction leads to enhanced inflammasome activation, TLR cross-talk, and immune exhaustion, producing a fibrogenic liver phenotype than ALD or MASLD alone [33].

Collectively, the inflammatory response in ALD represents a dynamic interplay between oxidative stress, innate immune activation, and adaptive immune dysregulation. Understanding these interactions, particularly in the context of overlapping metabolic risk, is crucial for the development of targeted immunotherapies in both ALD and MetALD.

### 2.4. Cellular and Molecular Consequences of Oxidative Injury

The accumulation of oxidative stress leads to widespread hepatocellular damage and dysfunction. Mitochondrial injury is central to this process and results in swelling, membrane depolarization, and cytochrome c release, initiating apoptotic signaling [35]. Concurrently, oxidative stress induces endoplasmic reticulum stress, which triggers the unfolded protein response and contributes to hepatocyte apoptosis or pyroptosis [36].

DNA damage is another critical consequence, marked by the presence of increased PARP1 activity. These changes disrupt genomic integrity and cell cycle regulation [37]. Protein carbonylation and formation of advanced oxidation protein products further compromise cellular function and contribute to the pro-inflammatory microenvironment of the liver [38].

Oxidative stress also activates hepatic non-parenchymal cells. Kupffer cells and recruited monocytes respond to damage-associated molecular patterns and acetaldehyde adducts by releasing pro-inflammatory cytokines. The cytokines promote hepatocyte injury and stimulate HSC activation, leading to collagen deposition and fibrosis [39].

### 2.5. Systemic and Neurological Perspectives

The consequences of alcohol-induced oxidative stress extend beyond the liver. Systemic circulation of ROS, cytokines, and acetaldehyde adducts has been implicated in cardiovascular dysfunction, immune dysregulation, and neuroinflammation [40]. In the central nervous system, oxidative stress contributes to blood–brain barrier disruption and microglial activation, linking chronic alcohol intake to cognitive impairment and increased cerebrovascular risk [41].

Despite the clear role of oxidative stress in the pathogenesis of ALD, clinical trials using general antioxidants have yielded mixed results. Several agents have shown some benefit in experimental models but limited success in human studies [42,43,44]. The detailed therapeutic options would be addressed in the section of therapeutic strategies.

In conclusion, alcohol metabolism initiates a complex cascade of oxidative stress and redox imbalance that plays a central role in liver injury and systemic complications. A more refined understanding of these mechanisms might enable the development of precision therapies, particularly in the emerging context of MetALD, where alcohol-related damage could affect liver with metabolic dysfunction to accelerate disease progression.

## 3. From Oxidative Stress to the Gut–Liver–Brain Axis: Emerging Insights into ALD and MetALD

### 3.1. Gut Barrier Dysfunction and Microbiome–Immune Crosstalk

The gastrointestinal tract plays a pivotal role in the pathogenesis of ALD through its dynamic interaction with the liver via the gut–liver axis. Chronic alcohol consumption first induces microbial dysbiosis, then disrupts epithelial tight junctions and mucus defense, and finally increases gut permeability, all of which present in the translocation of bacterial components and metabolites into the portal circulation. This alteration in gut integrity serves as a major source of immune activation in the liver [45].

Alcohol exerts its harmful effects on the gut through multiple mechanisms. Ethanol and its metabolite acetaldehyde directly impair tight junction proteins such as occludin, claudins, and zonula occludens-1, weakening the paracellular barrier between enterocytes. Concurrently, alcohol alters intestinal epithelial cell turnover, mucus secretion, and the antimicrobial peptide repertoire, further compromising mucosal defense. These structural and functional changes create a permissive environment for bacterial overgrowth and entry of PAMPs into the portal bloodstream [46].

Alcohol-induced dysbiosis is characterized by a reduction in commensal, beneficial microbial taxa, such as Lactobacillus and Bifidobacterium, and an overrepresentation of pathogenic bacteria including Enterobacteriaceae and Proteobacteria. These microbial shifts enhance the production of ethanol, acetaldehyde, ammonia, and other toxic metabolites, which further damage the intestinal barrier and modulate hepatic inflammation [47,48]. Additionally, dysbiosis changes the composition of microbial-derived short chain fatty acids and bile acids regulate not only Treg cells but also Th17 responses, macrophage and dendritic cell polarization, and intestinal or hepatocyte signaling pathways [49].

The gut–liver crosstalk also extends to the central nervous system through the gut–liver–brain axis. Increased intestinal permeability and microbial dysbiosis in ALD lead to systemic inflammation and neuroinflammation, partly mediated by circulating cytokines and microbial metabolites such as ammonia and endotoxin. These processes are implicated in cognitive dysfunction, mood disorders, and hepatic encephalopathy, highlighting the systemic consequences of gut barrier dysfunction [50,51].

Taken together, the gut–liver axis represents a crucial interface in ALD, mediating immune activation, fibrogenesis. Targeting this axis would be the key to investigate several intervention strategies that might improve both hepatic and extrahepatic outcomes.

### 3.2. Neuroinflammation and the Brain–Liver Axis in ALD

The central nervous system (CNS) is increasingly recognized as a critical target of alcohol-induced organ crosstalk, with evidence implicating the brain–liver axis in the pathogenesis of ALD. While the detrimental effects of chronic alcohol consumption on the liver are well established, recent studies have revealed that oxidative stress occurs first, followed by systemic inflammation, and then metabolic dysregulation, all extending pathological influence to the brain. This neuroimmune axis might contribute to cognitive impairment, mood disturbances, and increased cerebrovascular risk frequently observed in patients with ALD [52].

One of the most important mediators of this bidirectional communication is glutamate, the principal excitatory neurotransmitter in the brain [7,8]. Ethanol disrupts glutamate homeostasis by impairing astrocytic glutamate uptake and increasing extracellular glutamate concentrations, leading to excitotoxicity and neuronal injury [53]. Recent research suggests that glutamate signaling affects hepatic stellate cell activation and hepatocyte viability through xCT-mediated transportation [7]. These findings support the existence of a metabolic synapse between the liver and brain, whereby altered glutamatergic signaling in ALD might simultaneously drive hepatic fibrosis and neurotoxicity.

Another key pathological feature is disruption of the blood–brain barrier (BBB). Chronic alcohol exposure result in endothelial dysfunction and increased BBB permeability [54]. Several cytokines, which are abundantly produced in ALD, compromise BBB integrity and allow peripheral immune cells and microbial metabolites to access the CNS [55]. Acetaldehyde and ammonia, both elevated in advanced liver disease, exert direct neurotoxic effects that impair neurotransmission, mitochondrial function, and neural metabolism [56,57].

Microglial activation is a hallmark of alcohol-induced neuroinflammation. These resident immune cells of the CNS become chronically activated in response to peripheral and central inflammatory signals. Activated microglia produce ROS, nitric oxide, and cytokines, creating a feed-forward loop of neuroinflammation and neuronal injury. Additionally, alcohol alters microglial polarization, shifting them toward a proinflammatory phenotype while reducing anti-inflammatory responses. This imbalance contributes to impaired synaptic remodeling and cognitive decline [58].

Emerging evidence also suggests cerebrovascular involvement in ALD. Chronic alcohol use increases the risk of both ischemic and hemorrhagic stroke, partly through endothelial dysfunction, platelet activation, and hypertension [59]. Oxidative stress derived from liver injury would contribute to small vessel disease and impaired cerebral perfusion [60]. Furthermore, alcohol-induced disruption of lipid metabolism and insulin signaling might exacerbate neurodegeneration through mechanisms with those seen in Alzheimer’s disease and vascular cognitive impairment [61].

In summary, the brain–liver axis represents a critical component of ALD. Through BBB disruption and microglial activation, alcohol-induced inflammation and hepatic dysfunction converge to impair CNS integrity. Understanding this bidirectional communication opens new avenues for intervention that span beyond the liver and address the broader neuroimmune consequences of chronic alcohol use.

### 3.3. MetALD: A Converging Pathophysiologic Entity

The recent reclassification of liver diseases associated with metabolic dysfunction and alcohol intake has led to the emergence of a new clinical entity, MetALD. This term describes individuals who meet criteria for MASLD, such as obesity, insulin resistance, hyperlipidemia, or hypertension, while also consuming clinically significant amounts of alcohol. Rather than viewing ALD and MASLD as mutually exclusive, MetALD acknowledges the overlapping pathophysiologic contributions of both alcohol and metabolic risk factors in liver injury [11].

This intersectional framework challenges the traditional dichotomy of ALD versus MASLD and reflects the growing recognition that metabolic comorbidities are highly prevalent among individuals with alcohol use. In fact, a substantial proportion of ALD patients are overweight or obese, have impaired glucose tolerance, or exhibit features of metabolic syndrome, factors that independently worsen hepatic inflammation and fibrosis [62]. Conversely, many individuals previously classified as having MASLD consume alcohol at moderate-to-high levels, further complicating disease classification and therapeutic decision-making [63].

Pathophysiologically, MetALD represents a state of synergistic liver injury driven by both alcohol-induced and metabolic stress. Both pathways are characterized by oxidative stress, mitochondrial dysfunction, endoplasmic reticulum stress, and activation of innate immunity [64]. Alcohol enhances lipogenesis and suppresses fatty acid oxidation via changes in the NAD^+^/NADH ratio, while also promoting gut-derived endotoxemia and inflammatory cytokine release [65]. These processes are also central to MASLD progression, suggesting that MetALD is a “double-hit” condition in which alcohol and metabolic factors converge to accelerate fibrogenesis [64].

Recent large-scale cohort studies have demonstrated that individuals with MetALD are at a substantially higher risk of developing liver and gastrointestinal cancers, including esophageal and colorectal cancer, compared to those without steatotic liver disease [66]. Additionally, MetALD is associated with an increased risk of cardiovascular disease, highlighting the additive harmful effect of alcohol consumption in the presence of metabolic dysfunction [67]. These findings present the need for dedicated clinical attention and tailored management strategies for the MetALD population.

The recognition of MetALD represents a paradigm shift toward a more integrated view of liver disease, moving beyond simplistic categorizations based on alcohol use. As research evolves, it will be crucial to establish diagnostic criteria, incorporate MetALD populations into clinical trials, and develop therapies that address the dual burden of alcohol- and metabolism-related liver injury. Bridging this gap would be essential for advancing precision medicine in hepatology and improving outcomes for a growing population of patients with overlapping risk profiles.

Figure 1 illustrates how oxidative stress affects the gut-brain-liver axis in ALD and MetALD.

## 4. Neurologic and Metabolic Dimensions of ALD

ALD and its emerging phenotype, MetALD, are increasingly recognized as multisystemic disorders, with significant crosstalk between hepatic, neurologic, and metabolic pathways. This multidimensional pathophysiology extends beyond hepatocyte injury and is induced by chronic oxidative stress, systemic inflammation, and dysregulated inter-organ communication [68].

One of the most striking features of ALD is the involvement of the CNS. Ethanol and its metabolites, particularly acetaldehyde and ammonia, directly compromise the BBB, leading to microglial activation, and synaptic dysfunction [56,57]. These effects would be exacerbated by proinflammatory cytokines originating from hepatic immune activation [69]. Microglia, once activated, initiating a neuroinflammatory state that contributes to the cognitive decline, emotional dysregulation, and increased cerebrovascular risk commonly observed in ALD and advanced liver disease [69].

Concurrently, the liver is not an isolated target. The systemic burden of oxidative stress affects endothelial cells, skeletal muscle, adipose tissue, and pancreatic beta cells, addressing a cascade of metabolic disturbances. Insulin resistance, lipid dysregulation, and sarcopenia are frequently observed in ALD and especially in MetALD [70]. These features are active contributors to disease progression. Hyperinsulinemia and free fatty acid release sensitize hepatocytes to oxidative injury and prime immune cells for accelerated inflammatory responses [71]. This bidirectional amplification loop between liver and metabolic organs defines the severity of fibrosis and therapeutic responsiveness.

Moreover, the gut–liver–brain axis serves as a crucial point of pathology. Alcohol disrupts the intestinal barrier, allowing for microbial translocation and endotoxemia, which in turn activate hepatic and neural inflammatory pathways. Dysbiosis alters bile acid signaling and short-chain fatty acid production, with downstream effects on energy homeostasis, immune regulation, and even behavior and cognition [47]. In MetALD, the presence of metabolic syndrome augments this dysregulation, with increased systemic inflammation and altered gut hormone signaling further aggravating both hepatic and neurologic integrity [33].

These findings collectively suggest that ALD and MetALD are not confined to the liver but represent integrated disorders of oxidative and immunometabolic stress. Effective treatment strategies would account for this complexity. Precision targeting of redox imbalance, immune activation, and gut–brain–liver interactions offers a promising frontier but requires further mechanistic research and clinical validation. Understanding the interconnected nature of liver, brain, and metabolism will be essential to address the full clinical spectrum and improve patient outcomes.

## 5. Therapeutic Strategies Targeting Oxidative Stress and Neuroimmune Pathways

### 5.1. Antioxidant Therapies

Oxidative stress is central to the pathogenesis of ALD and MetALD, making antioxidant strategies an attractive therapeutic approach. However, most general antioxidants have produced disappointing clinical results. N-acetylcysteine, known for replenishing glutathione and directly scavenging free radicals, has shown benefit when combined with corticosteroids in acute alcoholic hepatitis but has not consistently improved long-term outcomes [43,72]. Other compounds, including S-adenosylmethionine, vitamin E, and zinc, have shown antioxidant activity in preclinical studies, but their translation into clinical practice has been limited by modest efficacy and unclear patient selection criteria [44,73].

More targeted approaches, such as mitochondrial-specific antioxidants or agents that restore NAD^+^ levels, might play a role by modulating the redox environment at critical cellular sites [74]. However, these remain investigational, and clinical trials are required to validate their therapeutic potential in ALD and MetALD.

### 5.2. Immune Modulating Approaches

Immune dysregulation driven by oxidative stress is another representative characteristic of ALD. Current immunosuppressive therapies show variable response rates, and non-responders often have poor prognoses. Moreover, immunosuppression-related complications limit their clinical utility in many patients [75].

Alternative immune-modulating strategies have focused on pro-inflammatory cytokines such as TNF-α and IL-1β. Nonetheless, clinical trials using agents including infliximab and anakinra have failed to demonstrate clinical benefit and were halted because of safety concerns [76,77]. More promising are approaches targeting pattern recognition receptors and intracellular signaling cascades, including TLR4 inhibitors, c-Jun N-terminal kinase/apoptosis signal-regulating kinase inhibitors, and inflammasome modulators [78,79]. While several agents are in preclinical or early clinical stages, further development is needed to ensure safety and identify appropriate patient subsets.

### 5.3. Targeting the Gut-Liver-Brain Axis

Alcohol-induced disruption of the intestinal barrier and dysbiosis contributes to systemic inflammation and neuroinflammation. Increased gut permeability allows for translocation of endotoxins like LPS, which activate hepatic and microglial TLR4 signaling, promoting both liver injury and cognitive dysfunction [80].

Therapies aimed at restoring gut-liver homeostasis are being actively investigated. Probiotics and prebiotics have shown promise in reducing endotoxemia and inflammation in small trials [81]. Fecal microbiota transplantation (FMT) is also being explored as a potential strategy in severe ALD [82]. On the neurocognitive front, agents such as minocycline and glutamate modulators are being studied for their neuroprotective effects in alcohol-related brain injury [68,83]. These gut- and brain-targeted therapies could eventually play a supportive role in managing systemic consequences of alcohol-induced oxidative stress.

Despite the clinical relevance, therapeutic strategies targeting neuroinflammation in ALD remain limited. Current management of hepatic encephalopathy focuses on ammonia-lowering agents such as lactulose and rifaximin, but does not directly address underlying neuroimmune dysregulation [84]. Experimental approaches including N-methyl-D-aspartate receptor antagonists, anti-cytokine therapies, and gut microbiome modulation are under investigation and might break the pathological connection between liver dysfunction and brain injury [85].

### 5.4. Challenges and Future Directions in MetALD Treatment

MetALD presents a unique therapeutic challenge, as patients exhibit both alcohol-induced and metabolic injuries. Unfortunately, they are often excluded from clinical trials targeting either ALD or MASLD. This has resulted in a lack of evidence to guide treatment in this group, despite data showing that MetALD patients have worse outcomes than those with ALD or MASLD alone.

Given this dual pathology, combination therapies may be required. Agents under investigation for MASLD, such as FXR agonists (obeticholic acid), fibroblast growth factor 21 analogs (pegbelfermin), or glucagon-like peptide-1 receptor agonists (semaglutide), might offer metabolic benefit, but their safety and efficacy in the context of alcohol use remains uncertain [86]. Future clinical trials should consider including MetALD populations and stratifying by alcohol intake, metabolic status, and inflammatory signatures to develop more personalized therapies.

Precision medicine approaches, integrating multi-omic profiling, lifestyle assessment, and real-time biomarkers of alcohol intake and oxidative stress, would ultimately be necessary to optimize treatment in this heterogenous population.

Table 1 summarizes pathophysiologic pathways and therapeutic targets in ALD and MetALD with neurologic dimensions.

## 6. Conclusions

ALD is a multifaceted disorder that extends well beyond the liver. While traditional frameworks focused on the quantity and duration of alcohol intake, recent findings present the central role of oxidative stress, immunometabolic dysregulation, and inter-organ communication, particularly involving the brain and metabolic systems. The emergence of MetALD further complicates the clinical landscape, revealing the need for more integrative diagnostic and therapeutic strategies.

Oxidative stress is a pivotal mechanism driving hepatocellular injury in ALD. Ethanol metabolism through ADH, ALDH, and CYP2E1 generates reactive oxygen and nitrogen species that overwhelm antioxidant defenses, damage mitochondria, and initiate inflammatory cascades. These processes are exacerbated by coexisting metabolic dysfunction, such as insulin resistance, obesity, and dyslipidemia. The result is more severe liver injury, accelerated fibrosis, and higher risk of complications such as HCC.

Importantly, the systemic impact of alcohol-induced oxidative stress is increasingly recognized. Circulating inflammatory mediators, acetaldehyde adducts, and oxidative byproducts affect vascular endothelium, immune cells, and the CNS. In the brain, chronic alcohol exposure leads to neuroinflammation, glutamate dysregulation, and cognitive decline. This highlights the significance of the gut–liver–brain axis in the pathogenesis of both hepatic and neurologic complications.

MetALD represents a convergence of two major public health challenges: alcohol misuse and metabolic disease. Patients with MetALD exhibit overlapping pathologies of ALD and MASLD, yet respond poorly to current treatments. Corticosteroids, commonly used for severe alcoholic hepatitis, are often less effective in the presence of metabolic comorbidities. Likewise, emerging metabolic-targeted therapies for MASLD might be suboptimal in patients with ongoing alcohol intake. These observations call for a rethinking of therapeutic strategies that account for the dual burden of alcohol and metabolic stress.

Despite advances in refining the pathogenesis of ALD, therapeutic development remains limited. Antioxidants such as N-acetylcysteine and vitamin E have shown inconsistent results in clinical trials. This emphasizes the need for more targeted interventions, including mitochondrial antioxidants, NAD^+^ boosters, or inhibitors of redox-sensitive signaling pathways. Additionally, future studies should evaluate investigational drugs in MetALD-specific cohorts to ensure efficacy in real-world populations.

Clinically, a revised diagnostic framework is required, one that acknowledges the spectrum between ALD and MASLD, incorporates metabolic risk stratification, and utilizes biomarkers reflective of oxidative stress and neuroinflammation. Multidisciplinary care and cure involving hepatology, endocrinology, and neurology would be essential to manage the multisystem nature of ALD and MetALD.

In summary, ALD is not limited to the liver. It is a systemic disorder driven by oxidative stress and shaped by complex interactions between metabolic, neurologic, and immune pathways. As the burden of MetALD rises globally, future research and clinical practice would evolve to reflect this multidimensional reality. Addressing oxidative stress at the crossroads of liver, brain, and metabolism might be key to improving outcomes in this challenging and increasingly prevalent disease entity.

## Figures and Tables

**Figure 1 antioxidants-14-01196-f001:**
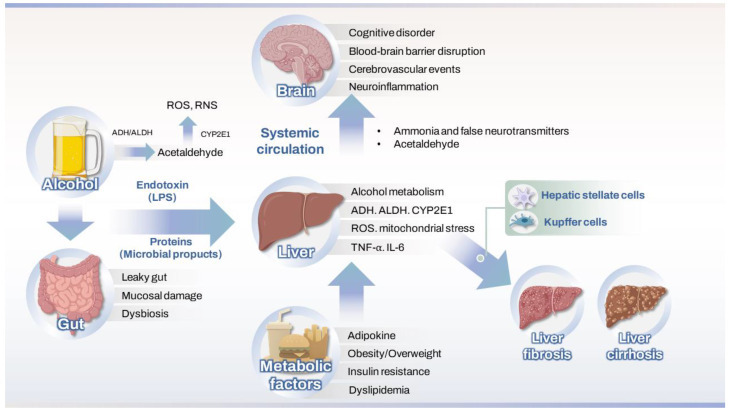
Mechanisms of oxidative stress affecting the gut–liver–brain axis in ALD and MetALD4. Therapeutic strategies targeting oxidative stress and neuroimmune pathways. ADH, alcohol dehydrogenase; ALDH, aldehyde dehydrogenase; CYP2E1, cytochrome P450 2E1; ROS, reactive oxygen species; RNS, reactive nitrogen species; LPS, lipopolysaccharide; BBB, blood–brain barrier; TNR, tumor necrosis factor.

**Table 1 antioxidants-14-01196-t001:** Pathophysiologic pathways and therapeutic targets in ALD and MetALD.

Mechanism	Mediators	Consequences	Potential Treatments
Ethanol metabolism (ADH, ALDH, CYP2E1, MEOS)	Acetaldehyde, NAD^+^/NADH imbalance, CYP2E1-induced ROS [13,14,15]	Hepatocyte injury, mitochondrial dysfunction, steatosis [16,19]	Vitamin E nanoemulsion (CYP2E1 inhibitor), NAD^+^ boosters [16,25]
Oxidative stress (ROS, RNS)	Superoxide, nitric oxide [20,21]	Lipid peroxidation, DNA/protein adducts, systemic oxidative burden [23,40]	Mitochondria-targeted antioxidants, S-adenosylmethionine, N-acetylcysteine [42,43,44]
Inflammatory signaling (NF-κB, JNK/ASK1, TLR4)	TNF-α, IL-1β, IL-6, inflammasome components [26,39]	Kupffer cell activation, HSC-driven fibrosis, systemic inflammation, microglial activation [27,58]	TLR4 antagonists, JNK/ASK1 inhibitors, IL-1 receptor antagonists [78,79]
Gut–liver–brain axis disruption	Acetaldehyde, endotoxin, dysbiosis [45,46]	Increased intestinal permeability, hepatic inflammation, systemic cytokines, neuroinflammation [50,52]	Probiotics, prebiotics, FMT, FXR agonists [82,87]
Brain–liver crosstalk	Glutamate excitotoxicity, BBB disruption, microglial ROS, ammonia [7,8,55,56]	Cognitive impairment, cerebrovascular dysfunction, neurodegeneration, hepatic encephalopathy [52,55,60]	NMDA receptor antagonists, microbiome-targeted interventions [80,85]
Metabolic insult (MetALD)	Hyperinsulinemia, free fatty acids, adipokines [33,62]	Accelerated fibrosis, sarcopenia, cardiometabolic complications, worsening CNS injury [33,66,67]	GLP-1 receptor agonists, FGF21 analogs, combination strategies [86]

ALD, alcohol-associated liver disease; MetALD, metabolic dysfunction-associated steatotic liver disease and increased alcohol intake; ADH, alcohol dehydrogenase; ALDH, aldehyde dehydrogenase; CYP2E1, cytochrome P450 2E1; MEOS, microsomal ethanol-oxidizing system; NAD^+^, nicotinamide adenine dinucleotide; ROS, reactive oxygen species; RNS, reactive nitrogen species; NF-κB, nuclear factor kappa-light-chain-enhancer of activated B cells; JNK, c-Jun N-terminal kinase; ASK1, apoptosis signal-regulating kinase 1; TLR4, Toll-like receptor 4; TNF-α, tumor necrosis factor-alpha; IL, interleukin; HSC, hepatic stellate cell; FMT, fecal microbiota transplantation; FXR, farnesoid X receptor; BBB, blood–brain barrier; NMDA, N-methyl-D-aspartate; CNS, central nervous system; GLP-1, glucagon-like peptide-1; FGF21, fibroblast growth factor 21.

## Data Availability

Data is contained within the article.

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
