# Peer review of "Alcohol-Induced Oxidative Stress and Gut–Liver–Brain Crosstalk: Expanding the Paradigm from ALD to MetALD"

_antioxidants, 2025, doi:10.3390/antiox14101196_

Round 1

Reviewer 1 Report

The review recapitulates the pernicious effects of alcohol consumption mainly in liver but with the interactions to gut and brain. Which will end in the definition of a new term to define the alcohol interactions with the metabolic disease. It is a nice review, but the organization of the sections is not clear, or the titles don't correspond well to what is inside. I will explain with some examples:

  1. Alcohol metabolism and oxidative stress in liver: I would do something very general about liver.

2.4. Systemic, neurological, and therapeutic perspectives: I would remove the systemic and neurological topics and keep just the therapeutic perspectives.

3.1. Immune activation and inflammatory pathways: this to the liver section.

  1. From oxidative stress to brain-liver axis: Emerging insights into ALD 151 and MetALD: here it is also described the gut, so it can be interesting to include it in the title.

As a suggestion, I think the 4 and 5 points should have the opposite order, first the 5 and then the 4.

-There is a duplication in table 1 tittle,

-Some of the abbreviations are missing, for instance TGFb, please check them all.

Author Response

The review recapitulates the pernicious effects of alcohol consumption mainly in liver but with the interactions to gut and brain. Which will end in the definition of a new term to define the alcohol interactions with the metabolic disease. It is a nice review, but the organization of the sections is not clear, or the titles don't correspond well to what is inside. I will explain with some examples:

Alcohol metabolism and oxidative stress in liver: I would do something very general about liver.

Answer: We thank the reviewer for this valuable suggestion. We have revised the section title to provide a broader scope that covers hepatic mechanisms in general, not limited to oxidative stress alone.

Section 2 is now titled.

Alcohol metabolism and liver pathophysiology

2.4. Systemic, neurological, and therapeutic perspectives: I would remove the systemic and neurological topics and keep just the therapeutic perspectives.

Answer: We sincerely appreciate the reviewer’s thoughtful observation. We initially considered focusing Section 2.4. on therapeutic perspectives only. However, as other reviewers also pointed out, treatment aspects are already extensively covered in the later section devoted to therapeutic strategies. To improve clarity and avoid redundancy, we have therefore modified Section 2.4 to provide only a brief mention of therapeutic options and focused the main discussion of treatments in Section 5. Accordingly, Section 2.4 has been retitled “Systemic and neurological perspectives” to better reflect its content. We believe this restructuring improves the logical flow and reduces overlap across the manuscript.

3.1. Immune activation and inflammatory pathways: this to the liver section.

Answer: We sincerely thank the reviewer for this valuable suggestion. According to the comment, the subsection on immune activation and inflammatory pathways has been relocated from the former Section 3.1 to Section 2.3, thereby consolidating all liver-related mechanisms (ethanol metabolism, oxidative stress, and immune activation) within the same section.

From oxidative stress to brain-liver axis: Emerging insights into ALD 151 and MetALD: here it is also described the gut, so it can be interesting to include it in the title.

Answer: We fully agree that the gut is a critical component. We have revised the section title to reflect the inclusion of the gut along with the liver and brain.

Section 3 is now titled.

From oxidative stress to the gut–liver–brain axis: Emerging insights into ALD and MetALD

As a suggestion, I think the 4 and 5 points should have the opposite order, first the 5 and then the 4.

Answer: We thank the reviewer for this thoughtful suggestion. We have reordered the sections so that neurologic and metabolic dimensions are presented before therapeutic strategies, which provides a more logical progression from mechanisms to outcomes and then treatment.

-There is a duplication in table 1 tittle,

Answer: We thank the reviewer for noting this formatting issue. The duplication in the title of Table 1 has been corrected. The revised title is now presented in a concise format.

Table 1 is now titled.

Table 1. Pathophysiologic pathways and therapeutic targets in ALD and MetALD

-Some of the abbreviations are missing, for instance TGFb, please check them all.

Answer: We thank the reviewer for carefully pointing this out. We have re-checked the entire manuscript and ensured that all abbreviations are consistently listed and defined. Missing items, including TGF-β, have been added.

Reviewer 2 Report

Risky alcohol consumption and its severe health consequences, including alcohol-associated liver disease (ALD) and newly introduced metabolic dysfunction and alcohol associated/related liver disease (MetALD) create a global health problem. Also in East Asia, alcohol use is on the rise, and as reported, the Republic of Korea has the highest annual alcohol consumption per capita among all countries within the Asia Pacific region [Monzavi SM, et al. Alcohol Related Disorders in the Asia Pacific Region: Prevalence, Health Consequences and Impacts on the Nations. Asia Pacific Journal of Medical Toxicology. 2015]. Therefore, the subject of the present paper is relevant. However, numerous issues should be addressed.

The main problem with the manuscript is that the authors describe the ALD and MetALD theories but do not present any detailed results from clinical or experimental studies to support and confirm their statements. It looks like the manuscript sharply keeps to the scope of the journal, indicating oxidative stress in every part of it. More detailed information is needed. The other significant issue is the enormous repetition of information throughout the paper. Therefore, the manuscript requires major revision before it can be reconsidered for publication.

Line 38 “…transient psychological benefits…”- what do the authors mean?

Line 56-58 The authors claim that “Recent studies have highlighted the importance of liver-brain 56 interactions, where neurotransmitters such as glutamate and endocannabinoids modulate 57 hepatic stellate cell activation, inflammation, and fibrogenesis [8,9].”- but both cited reports indicate the local hepatic glutamate impact, not liver-brain interactions.

Line 65-69 The difference between ALD and MetALD should be described in detail. ALD is a distinct liver condition classified within the whole steatotic liver disease group, and MetALD differs from ALD based not only on the presence of metabolic risk factors, but also the quantity of alcohol consumed.

Line 164  “In the alcoholic liver disease…”- please use the recommended terminology, i.e., alcohol-related liver disease

Line 181 “Adipose tissue-derived cytokines such 180 as leptin accelerate inflammatory pathways…”- but not only leptin is engaged in hepatic inflammation

Therapeutic strategies are repeated throughout the whole manuscript, but there is a special part of the paper, which is  4. Therapeutic strategies… so please modify the text to avoid repetition in the treatment description

Line 202-2024 “Chronic alcohol consumption disrupts the intestinal epithelial barrier, increases permeability, and leads to microbial dysbiosis…”-  the sentence should be modified according to the sequence of events into: Chronic alcohol consumption leads to microbial dysbiosis (first), disrupts the intestinal epithelial barrier (second), and increases gut permeability (third)

Line 225-226 “…microbial-derived short-chain 225 fatty acids and bile acids, which are known to influence Treg cell development, intestinal 226 permeability, and hepatocyte signaling pathways…”

Why do authors mention only Tregs if many cell types are affected by SCFAs and BAs?

Line 249-250 “…recent studies have revealed that systemic inflammation, oxidative stress, and metabolic dysregulation initiated in the liver extend their pathological influence to the brain….”- …”-  the sentence should be modified according to the sequence of events into: recent studies have revealed that oxidative stress (first), systemic inflammation (second), and metabolic dysregulation (third) initiated in the liver extend their pathological influence to the brain…

Glutamate impact on metabolic dysregulation is repeated many times as follows lines: 56-61; 256-258, 291-292, 419-420; 462-465; also immune activation, systemic inflammation, corticosteroid treatment, etc.

Line 287 “NMDA”, please explain the abbreviations at their first use.

Figure 1 does not present only mechanisms of oxidative stress as indicated in the Figure title. Why is acetaldehyde set next to the glass with an alcoholic beverage? What proteins penetrate with endotoxin into the liver? Not only ammonia, but also other false transmitters penetrate through the blood-brain barrier. Neuroinflammation takes place in the brain, not in systemic circulation. No single hepatic stellate cell or Kupffer cell accounts for the development of liver disease (should be cells). Metabolic factors include adipokines (not adinokine), obesity/overweight, and dyslipidemia. Please explain the abbreviations below the figure.

Table 1: hepatic encephalopathy should be included in the consequences of brain-liver crosstalk and BBB disruption

My conclusion: Major revision of the manuscript is required. I do not recommend publishing the manuscript in its present form.

Author Response

Major comments

Risky alcohol consumption and its severe health consequences, including alcohol-associated liver disease (ALD) and newly introduced metabolic dysfunction and alcohol associated/related liver disease (MetALD) create a global health problem. Also in East Asia, alcohol use is on the rise, and as reported, the Republic of Korea has the highest annual alcohol consumption per capita among all countries within the Asia Pacific region [Monzavi SM, et al. Alcohol Related Disorders in the Asia Pacific Region: Prevalence, Health Consequences and Impacts on the Nations. Asia Pacific Journal of Medical Toxicology. 2015]. Therefore, the subject of the present paper is relevant. However, numerous issues should be addressed.

The main problem with the manuscript is that the authors describe the ALD and MetALD theories but do not present any detailed results from clinical or experimental studies to support and confirm their statements. It looks like the manuscript sharply keeps to the scope of the journal, indicating oxidative stress in every part of it. More detailed information is needed. The other significant issue is the enormous repetition of information throughout the paper. Therefore, the manuscript requires major revision before it can be reconsidered for publication.

Answer: We fully agree with the reviewer’s concerns and are grateful for highlighting this important point. In the revised version, we have detailed components with cited studies to specify each mechanistic statement. We have modified the text to avoid unnecessary repetition. For example, therapeutic strategies are now concentrated in Section 5, while mechanistic details are linked more directly to the supporting literature. We believe these changes have made the manuscript more concise and informative.

Detailed comments

Line 38 “…transient psychological benefits…”- what do the authors mean?

Answer: We appreciate the reviewer’s request for clarification. This phrase has been revised to specify that some individuals perceive alcohol to provide short-term mood elevation and social relaxation effects, while also emphasizing that these are temporary and do not reduce the harmful long-term consequences.

Below is modified Introduction section.

While moderate consumption might be used for short-term mood elevation and social relaxation effects, chronic and excessive alcohol intake remains a major global health concern.

Line 56-58 The authors claim that “Recent studies have highlighted the importance of liver-brain 56 interactions, where neurotransmitters such as glutamate and endocannabinoids modulate 57 hepatic stellate cell activation, inflammation, and fibrogenesis [8,9].”- but both cited reports indicate the local hepatic glutamate impact, not liver-brain interactions.

Answer: We thank the reviewer for pointing out this important distinction. We have modified the text to clarify that the cited studies describe hepatic glutamate signaling, and that the implication for brain–liver crosstalk is indirect. This change prevents over-interpretation of the cited data and strengthens accuracy.

Below is modified Introduction section.

Recent studies have highlighted hepatic glutamate signaling as hepatic stellate cell (HSC) activation and natural killer cell modulation, which might indirectly contribute to systemic inflammation and brain–liver crosstalk.

Line 65-69 The difference between ALD and MetALD should be described in detail. ALD is a distinct liver condition classified within the whole steatotic liver disease group, and MetALD differs from ALD based not only on the presence of metabolic risk factors, but also the quantity of alcohol consumed.

Answer: We agree that a clearer description is needed and thank the reviewer for the suggestion. In the revised Introduction, we added a short subsection comparing ALD, MASLD, and MetALD, clearly stating alcohol thresholds and the role of metabolic dysfunction.

Below is modified Introduction section.

ALD refers to steatotic liver disease driven primarily by harmful alcohol use (defined as > 60 g/day in men or > 50–60 g/day in women, equivalent to > 420 g/week and > 350 g/week, respectively). MASLD refers to steatosis in the presence of metabolic dysfunction when alcohol use is below harmful thresholds (≤ 210 g/week for men and ≤ 140 g/week for women). MetALD describes patients meeting MASLD criteria while consuming alcohol above these MASLD thresholds but below harmful ranges (210–420 g/week for men; 140–350 g/week for women), reflecting a dual burden of metabolic dysfunction and alcohol use.

Line 164 “In the alcoholic liver disease…”- please use the recommended terminology, i.e., alcohol-related liver disease

Answer: We appreciate this correction and have carefully revised the terminology throughout the entire manuscript. All instances of “alcoholic liver disease” now read “alcohol-associated liver disease (ALD)”, which is consistent with current international recommendations and improves alignment with contemporary nomenclature.

Line 181 “Adipose tissue-derived cytokines such 180 as leptin accelerate inflammatory pathways…”- but not only leptin is engaged in hepatic inflammation

Answer: We thank the reviewer for drawing attention to this oversimplification. We have expanded the description to include other key adipokines such as adiponectin and resistin, as well as additional cytokines/chemokines implicated in hepatic inflammation.

Below is modified section “Immune activation and inflammatory pathways”

Adipose-derived mediators including leptin, adiponectin, resistin, and other cytokines/chemokines modulate Kupffer cell priming, HSC activation, and systemic inflammation.

Therapeutic strategies are repeated throughout the whole manuscript, but there is a special part of the paper, which is 4. Therapeutic strategies… so please modify the text to avoid repetition in the treatment description

Answer: We appreciate this valuable observation and have addressed it by consolidating therapeutic information into section of Therapeutic strategies. In earlier sections, we now provide only brief contextual mentions in Section 2.4 and refer the reader directly to Section for Therapeutic strategies.

Below is modified section 2.4.

Despite the clear role of oxidative stress in the pathogenesis of ALD, clinical trials using general antioxidants have yielded mixed results. Several agents have shown some benefit in experimental models but limited success in human studies. The detailed therapeutic options would be addressed in the section of therapeutic strategies.

Line 202-2024 “Chronic alcohol consumption disrupts the intestinal epithelial barrier, increases permeability, and leads to microbial dysbiosis…”-  the sentence should be modified according to the sequence of events into: Chronic alcohol consumption leads to microbial dysbiosis (first), disrupts the intestinal epithelial barrier (second), and increases gut permeability (third)

Answer: We thank the reviewer for the careful reading and agree with the suggested sequence. The text has been revised to indicate that chronic alcohol consumption first induces microbial dysbiosis, then disrupts the intestinal epithelial barrier, and finally increases gut permeability.

Below is modified section “Gut barrier dysfunction and microbiome–immune crosstalk”

Chronic alcohol consumption first induces microbial dysbiosis, then disrupts epithelial tight junctions and mucus defense, and finally increases gut permeability, all of which present in the translocation of bacterial components and metabolites into the portal circulation.

Line 225-226 “…microbial-derived short-chain 225 fatty acids and bile acids, which are known to influence Treg cell development, intestinal 226 permeability, and hepatocyte signaling pathways…”

Why do authors mention only Tregs if many cell types are affected by SCFAs and BAs?

Answer: We appreciate this insightful comment. The revised text now emphasizes that SCFAs and bile acids modulate not only Treg cells but also Th17 cells, macrophages, dendritic cells, and innate lymphoid cells.

Below is modified section “Gut barrier dysfunction and microbiome–immune crosstalk”

Additionally, dysbiosis changes the composition of microbial-derived short chain fatty acids and bile acids regulate not only Treg cells but also Th17 responses, macrophage and dendritic cell polarization, and intestinal or hepatocyte signaling pathways

Line 249-250 “…recent studies have revealed that systemic inflammation, oxidative stress, and metabolic dysregulation initiated in the liver extend their pathological influence to the brain….”- …”- the sentence should be modified according to the sequence of events into: recent studies have revealed that oxidative stress (first), systemic inflammation (second), and metabolic dysregulation (third) initiated in the liver extend their pathological influence to the brain…

Answer: We thank the reviewer for this precise suggestion. The sentence has been modified to state that oxidative stress occurs first, followed by systemic inflammation, and then metabolic dysregulation, all of which extend pathological influence to the brain.

Below is modified section “Neuroinflammation and the Brain–liver Axis in ALD”

While the detrimental effects of chronic alcohol consumption on the liver are well established, recent studies have revealed that oxidative stress occurs first, followed by systemic inflammation, and then metabolic dysregulation, all extending pathological influence to the brain

Glutamate impact on metabolic dysregulation is repeated many times as follows lines: 56-61; 256-258, 291-292, 419-420; 462-465; also immune activation, systemic inflammation, corticosteroid treatment, etc.

Answer: We acknowledge this important point and have carefully revised the text to eliminate redundancy. Glutamate signaling is now introduced in the Introduction and elaborated in Section “Neuroinflammation and the Brain–liver Axis in ALD”, with other mentions cross-referenced. Similarly, immune activation and systemic inflammation are discussed in detail in Section “Immune activation and inflammatory pathways”, and corticosteroid therapy is treated in Section 5, with deleting repeated addressed terms.

Line 287 “NMDA”, please explain the abbreviations at their first use.

Answer: We thank the reviewer for noticing this omission. The abbreviation has been fully spelled out as N-methyl-D-aspartate (NMDA) receptor at its first occurrence.

Below is modified section “Neuroinflammation and the Brain–liver Axis in ALD”

Experimental approaches including N-methyl-D-aspartate receptor antagonists, anti-cytokine therapies, and gut microbiome modulation are under investigation and might break the pathological connection between liver dysfunction and brain injur

Figure 1 does not present only mechanisms of oxidative stress as indicated in the Figure title. Why is acetaldehyde set next to the glass with an alcoholic beverage? What proteins penetrate with endotoxin into the liver? Not only ammonia, but also other false transmitters penetrate through the blood-brain barrier. Neuroinflammation takes place in the brain, not in systemic circulation. No single hepatic stellate cell or Kupffer cell accounts for the development of liver disease (should be cells). Metabolic factors include adipokines (not adinokine), obesity/overweight, and dyslipidemia. Please explain the abbreviations below the figure.

Answer: We are grateful for the detailed feedback on Figure 1. The figure has been revised with a more appropriate title, corrected representation of acetaldehyde, clarified description of gut-derived factors (LPS and microbial products), recognition of multiple false neurotransmitters beyond ammonia, and corrected terminology (“adipokines”). We also added a comprehensive legend that defines all abbreviations.

Table 1: hepatic encephalopathy should be included in the consequences of brain-liver crosstalk and BBB disruption

Answer: We thank the reviewer for this suggestion, which indeed strengthens the table. Hepatic encephalopathy has been added under the “brain–liver crosstalk” consequences column, with cognitive impairment and cerebrovascular dysfunction.

My conclusion: Major revision of the manuscript is required. I do not recommend publishing the manuscript in its present form.

Round 2

Reviewer 1 Report

Thank you for following my suggestions

Thank you for following my suggestions

Author Response

Thank you for following my suggestions.

Answer: We sincerely thank the reviewer for the helpful suggestions. We are pleased that the revised version has addressed the concerns raised.

Reviewer 2 Report

My suggestions were not followed. 

There are still some repetitions throughout the paper, including ethanol metabolism (lines 47-50 and 92-106), amounts of alcohol consumption (lines 43-44 and 307), pro-inflammatory cytokines (lines 130-131, 184-185, 223-224, 271) etc. I recommended moving all therapeutic strategies into one section, but the authors have not followed my recommendation (see lines 157-165, 242-248, 290-295) 
The sentences (lines 403-405) are unclear- probably some of the text was lost because the sentences did not fit the context. 

Major revision is needed.

Author Response

Response to Reviewer

We sincerely thank the reviewer for the constructive comments, which have greatly improved the clarity and organization of our manuscript. We also apologize that our initial revision did not fully address several of the reviewer’s important suggestions. In the present revision, we have carefully reconsidered all comments. Below, we provide a detailed point-by-point response.

Comment 1. There are still some repetitions throughout the paper, including ethanol metabolism (lines 47–50 and 92–106), amounts of alcohol consumption (lines 43–44 and 307), pro-inflammatory cytokines (lines 130–131, 184–185, 223–224, 271) etc.

Answer: We appreciate the reviewer’s careful observation. We have revised the manuscript to eliminate redundancy. Specifically:

  1. All descriptions of ethanol metabolism have been consolidated into Section 2.
  2. The information regarding the amount of alcohol consumption is now described only in the Introduction, and removed from other sections.
  3. All mentions of pro-inflammatory cytokines have been reorganized and consolidated into Section 2.3, thereby avoiding repetition.

Comment 2. I recommended moving all therapeutic strategies into one section, but the authors have not followed my recommendation (see lines 157–165, 242–248, 290–295).

Answer: We sincerely apologize for not fully addressing this important recommendation in our previous revision. In the revised manuscript, all therapeutic strategies have now been reorganized and integrated into a single section (Section 5). We have removed therapeutic content from other sections to improve flow, as suggested.

Comment 3. The sentences (lines 403–405) are unclear—probably some of the text was lost because the sentences did not fit the context.

Answer: Thank you for this valuable comment. We recognized that the previous version was ambiguous.

To improve clarity, we have revised this passage as follows.

Immune dysregulation driven by oxidative stress is another representative characteristic of ALD. Current immunosuppressive therapies show variable response rates, and non-responders often have poor prognoses. Moreover, immunosuppression-related complications limit their clinical utility in many patients.

Alternative immune-modulating strategies have focused on pro-inflammatory cytokines such as TNF-α and IL-1β. Nonetheless, clinical trials using agents including infliximab and anakinra have failed to demonstrate clinical benefit and were halted because of safety concerns.

We believe this revision resolves the reviewer’s concern by clearly stating the therapeutic context, specifying the agents, and ensuring logical flow between sentences.